# Spatiotemporal Analysis of Black Carbon Sources: Case of Santiago, Chile, under SARS-CoV-2 Lockdowns

**DOI:** 10.3390/ijerph192417064

**Published:** 2022-12-19

**Authors:** Carla Adasme, Ana María Villalobos, Héctor Jorquera

**Affiliations:** 1Departamento de Ingeniería Química y Bioprocesos, Pontificia Universidad Católica de Chile, Santiago 7820436, Chile; 2Centro de Desarrollo Urbano Sustentable (CEDEUS), Los Navegantes 1963, Providencia, Santiago 7520246, Chile

**Keywords:** black carbon, aethalometer model, spatiotemporal patterns, fuzzy clustering, FUSTA

## Abstract

Background: The SARS-CoV-2 pandemic has temporarily decreased black carbon emissions worldwide. The use of multi-wavelength aethalometers provides a quantitative apportionment of black carbon (BC) from fossil fuels (BC_ff_) and wood-burning sources (BC_wb_). However, this apportionment is aggregated: local and non-local BC sources are lumped together in the aethalometer results. Methods: We propose a spatiotemporal analysis of BC results along with meteorological data, using a fuzzy clustering approach, to resolve local and non-local BC contributions. We apply this methodology to BC measurements taken at an urban site in Santiago, Chile, from March through December 2020, including lockdown periods of different intensities. Results: BC_ff_ accounts for 85% of total BC; there was up to an 80% reduction in total BC during the most restrictive lockdowns (April–June); the reduction was 40–50% in periods with less restrictive lockdowns. The new methodology can apportion BC_ff_ and BC_wb_ into local and non-local contributions; local traffic (wood burning) sources account for 66% (86%) of BC_ff_ (BC_wb_). Conclusions: The intensive lockdowns brought down ambient BC across the city. The proposed fuzzy clustering methodology can resolve local and non-local contributions to BC in urban zones.

## 1. Introduction

Black carbon (BC) is one of the components of fine respirable particle matter (PM_2.5_); it comes from the incomplete combustion of fossil fuels and biomass. Exposure to BC has been linked to short-term [1,2] and long-term [3,4,5,6,7] health effects, but its regulation is indirect through the regulation of ambient PM_2.5_. Recently, the World Health Organization has updated its air quality guidelines [8], setting an annual average of PM_2.5_ of 5 μg/m^3^, which means that long-term BC is implicitly recommended to be well below that guideline since BC is usually below 20% of the total PM_2.5_.

BC has been traditionally measured offline using thermal-optical methods applied to filter samples [9,10]; these results are reported as total BC in PM_2.5_ [11]. More recently, continuous instruments based on optical absorption at several wavelengths (from UV to IR) have been developed. These instruments (aethalometers) can apportion BC coming from fossil fuel (BC_ff_) and wood burning (BC_wb_) combustion because the BC emitted from those sources has a different wavelength dependence for that absorption [12]. This technological development has led to many studies worldwide that report that BC source apportionment in urban [13,14,15,16,17,18,19,20] and rural areas [21,22,23]. Despite this improvement, those apportionment results—on any given receptor site—report the total BC_ff_ (or BC_wb_) coming from local and non-local sources; for instance, regional wildfires may contribute to BC_wb_ as much as local sources, BC_ff_ may come from local and regional traffic sources, etc. Additional tools, like air quality models, have been used to resolve those local and non-local BC contributions [24]. Recently, aethalometers have been used to assess the changes in ambient BC_ff_ and BC_wb_ associated with urban lockdowns worldwide [25,26,27,28,29,30,31]. All these studies report significant decreases in ambient BC concentrations under those exceptional circumstances, with traffic sources being the largest contributors to those decreases, while BC_wb_ sometimes has not changed [29] or has even increased [32].

The purpose of this work is twofold: (a) report ambient BC_ff_ and BC_wb_ concentrations for the very first time in Santiago, Chile, and estimate the reductions in ambient BC concentrations brought by lockdowns during SARS-CoV-2 pandemics, (b) apply a new methodology of spatiotemporal pattern recognition for estimating local and non-local contributions to ambient BC_ff_ and BC_wb_.

The new methodology is based on a fuzzy clustering algorithm applied to ambient BC_ff_ and BC_wb_ concentrations along with surface meteorological variables (wind speed and direction, air temperature). This methodology—named FUSTA (Fuzzy SpatioTemporal Apportionment)—splits ambient concentrations into several spatiotemporal patterns, each one corresponding to a contribution from one of the major emission sources [32]. This novel method generates a source apportionment for local and non-local BC sources without the need for air quality modeling applied to the city. The latter would require (a) an accurate emission inventory for BC_ff_ and BC_wb_, (b) the meteorological input fields should be accurate and capture the strong mixing layer seasonality over Santiago, and (c) the air quality model used should not have significant biases.

We find a reduction in total BC in Santiago during the lockdowns in 2020, from 40% to 80%, as compared with previous measurements in 2015; we also find that the FUSTA approach is a useful tool to resolve local and non-local sources of BC_ff_ and BC_wb_.

## 2. Materials and Methods

The methodology follows a sequential process, as shown in Figure 1. Below, we describe each of the methodological steps.

### 2.1. Ambient Measurement Campaign

The measurement campaign was carried out between 16 March 2020 and 2 January 2021. Lockdowns started on 27 March in the NE part of the city, and on 23 April, the SW sector of the city was added. Later on 15 May, a total lockdown was enacted until 27 July, followed by less restrictive lockdowns in the city until 30 November, when another rise in people infected forced the government to increase mobility restrictions again [33].

The monitoring was conducted using a multiwavelength aethalometer (MA200, San Francisco, CA, USA) measuring at five wavelengths: 375, 470, 538, 625, and 880 nm, corresponding to ultraviolet, blue, green, red and infrared, respectively. The monitoring site was chosen in a residential area located on the east border of the city (33.406° S, 70.512° W). Surface meteorological data were taken from a nearby site (33.377° S, 70.523° W), which corresponds to an air quality station (Las Condes) run by the Ministry of the Environment [34]. The location of the monitoring site was chosen on the east border of the city to capture the city’s pollution plume arriving at that site when daylight anabatic winds develop. That zone of the city has been studied before with ambient BC campaigns [24], so there was a baseline available to make comparisons with/without lockdowns.

The total BC signal recorded from the instrument was calibrated against the thermal optical transmittance method (TOT) NIOSH 5050 results applied to co-located PM_2.5_ samples taken on 47 mm quartz filters (Pallflex Tissuquartz 2500QAT-UP, Pall Life Sciences, Portsmouth, UK) using a minivol sampler (Super SASS, MetOne Instruments, Grants Pass, OR, USA); the BC TOT analysis was carried out at Chester LabNet (Tigard, OR, USA).

### 2.2. Aethalometer Data Analysis

Hourly averages of absorption coefficients (b_abs_) measured at 375 and 880 nm reported by the MA200 are used to compute the Absorption Ångström Exponent (AAE) according to [12]:AAE = −ln(b_abs_(375 nm)/b_abs_(880 nm))/ln(375/880)(1)

The histogram of hourly values of AAE is analyzed, and the 1st and 99th percentiles are identified with the Ångström exponents for fossil fuel (AAE_ff_) and wood burning (AAE_wb_), respectively [35]; the estimated values are AAE_ff_ = 0.7 and AAE_wb_ = 2.48. Appendix B shows how this estimation was carried out.

Next, the contributions BC_ff_ and BC_wb_ are computed as [12]:BC_ff_ = BC_total_·b_abs,ff_(880 nm)/b_abs_(880 nm)(2)
BC_wb_ = BC_total_·b_abs,wb_(880 nm)/b_abs_(880 nm)(3)
where BC_total_ is the total BC reading of the instrument at 880 nm, and the following expressions are used to estimate the absorption coefficients b_abs,ff_ and b_abs,wb_ [12]:b_abs,ff_(880 nm) = {b_abs_(375 nm) − b_abs_(880 nm)·(375/880)^−AAEff^}/{(375/880)^−AAEwb^ − (375/880)^−AAEff^}(4)
b_abs,wb_(880 nm) = {b_abs_(375 nm) − b_abs_(880 nm)·(375/880)^−AAEwb^}/{(375/880)^−AAEff^ − (375/880)^−AAEwb^}(5)

### 2.3. Spatiotemporal Data Analysis

In a previous publication [36], we used bivariate plots and k-means clustering of ambient PM_2.5_ and PM_10_, along with receptor model results, to estimate major sources contributing to ambient PM in urban areas; this methodology works best when one or two sources are the major contributors to ambient concentrations. However, this approach has two limitations: (i) the bivariate plots accept only pairs of meteorological variables to analyze ambient PM concentrations, (ii) the clustering technique is hard, that is, each hourly observation may belong to only one cluster (source). To improve the flexibility of that analysis, the meteorological input variables were increased to four: wind speed, wind direction, temperature, and pressure. But the key improvement is to use a fuzzy clustering algorithm, so each hourly observation may belong to more than one (fuzzy) cluster, using the probabilistic concept of cluster membership [37]. The proof of the concept of this new approach (denoted as FUSTA: FUzzy SpatioTemporal Apportionment) was developed for ambient SO_2_ in an industrial zone, where it was shown that spatiotemporal patterns obtained from FUSTA were like the ones obtained by air quality modeling of the major SO_2_ emission sources in the study zone [32]. This was the rationale for hypothesizing that FUSTA could resolve local and non-local sources of BC_ff_ and BC_wb_ because these are inert tracers of combustion sources, so they are only subject to atmospheric transport and deposition. Below we summarize the major steps needed to carry out such a methodology for the case of black carbon.

Data of BC_ff_ and BC_wb_ are log transformed to approach a normal distribution. Each of them is combined with air temperature and pressure and the Cartesian components of wind speed as in the case of bivariate plots [38]. These 5D databases are analyzed to find spatiotemporal patterns in BC_ff_ and BC_wb_ by using the algorithm FKM.ent.noise [39] available in the library fclust in the R environment [40]. The following optimization is carried out to find the centroids {C} and membership values U = {u_ij_} for the case of p fuzzy clusters sought:(6)$$\begin{array}{l} \underset{U,C}{min}J_{FKMNE}=\overset{n}{\underset{i=1}{\sum }}\overset{p}{\underset{k=1}{\sum }}u_{ik}\cdot ||x_{i}-c_{k}{||}^{2}+t\cdot \overset{n}{\underset{i=1}{\sum }}\overset{p}{\underset{k=1}{\sum }}u_{ik}\cdot log\left(u_{ik}\right)+\overset{n}{\underset{i=1}{\sum }}\delta^{2}{\left(1-\overset{p}{\underset{k=1}{\sum }}u_{ik}\right)}^{2}\\ s.t.\;u_{ik}\in \left[0,\;1\right];\;\overset{p+1}{\underset{k=1}{\sum }}u_{ik}=1\; \end{array}$$
where we use the default values t = 1 and δ = 1 in the above equation [39]. The very last term on the right-hand side of (6) stands for a noise cluster, that is, a subset of data that does not follow a regular pattern as the other *p* fuzzy clusters do [41]. This noise cluster includes outlier values or contributions from intermittent sources like a structural fire or a wildfire plume reaching the monitoring site, for instance.

Once the solution of (6) is found, the BC_ff_ estimated at time ‘i’ from Equations (2) and (4) is apportioned as follows:(7)$$BC_{ffi}=\overset{p+1}{\underset{k=1}{\sum }}BC_{ffi}\cdot u_{ik}=\overset{p}{\underset{k=1}{\sum }}BC_{ffi,\;k}+BC_{ffi,\;noise}\;\;$$
where $BC_{ffi,\;k}$ stands for the contribution of the *k*-th cluster (or source) to $BC_{ffi}$. A similar equation holds for BC_wbi_. Note that, by design, all those contributions are non-negative.

Since the results are 5D objects, we project the resulting fuzzy clusters using 3 different bivariate plots [38,42,43] in which wind direction is combined with wind speed, temperature, and pressure, respectively, to visualize the spatial distribution of fuzzy clusters found for each BC fraction. These graphs support the task of identifying each of the fuzzy clusters resolved by the FUSTA algorithm (6).

The database and all routines used in the data analysis and visualization are provided as Appendix A.

## 3. Results

### 3.1. Ambient Monitoring Results

#### 3.1.1. Absorption Ångström Exponents (AAE)

The following table lists the statistics for the estimated absorption Ångström exponents and estimated concentrations of BC_ff_, BC_wb_ and BC for the whole campaign.

Figure 2 shows the diel profiles of AAE for the austral summer and winter months. The winter mean value is significantly higher than the summer value (t = 11.9, *p*-value < 2.2 × 10^−16^), which suggests that wood-burning contributions to AAE increase in winter because of residential space heating in the city, a well-known source of ambient PM_2.5_ in Santiago [44].

Figure 3 shows a comparison of diel profiles of AAE for workdays and weekends. During weekends, the AAE mean value is significantly higher than in the case of workdays (t = 7.64, *p*-value = 2.8 × 10^−14^); this suggests a higher consumption of wood burning on weekends and thus the increase in AAE values.

#### 3.1.2. BC_ff_ and BC_wb_ Results

Figure 4 shows the time variability for BC_ff_ and BC_wb_ contributions estimated from the aethalometer model. BC_ff_ is the dominant contribution to total BC all year long; this contribution decreases over weekends, as expected from the traffic activity variability in the city. From Table 1, it follows that, on average, BC_ff_ accounts for 85% of total BC. Regarding BC_wb_ contribution (see Figure A3), it rises in winter months, as expected, and it does not decrease over weekends since it comes from residential sources. This contribution does not vanish in the spring and summer seasons; this is explained by wildfires and agricultural burning sources at the regional scale; they have been found in Santiago using receptor modeling of ambient PM_2.5_ combined with satellite images [45].

#### 3.1.3. Effect of SARS-CoV-2 Lockdowns on BC Concentrations

There is no continuous monitoring of ambient BC in Santiago. However, there was an ambient monitoring campaign that included BC measurements at nearby Las Condes station from December 2014 through July 2015, using an aethalometer (Magee Scientific, Berkeley, CA, USA, model AE33); that campaign results are reported in [24]. Table 2 below makes a comparison of monthly average BC values between that campaign and present results. The most intensive city lockdowns led up to an 80% of reduction in total BC (June 2020), and a 40% reduction has been estimated with fewer intensive lockdowns in December 2020 [33].

### 3.2. Spatiotemporal Analysis

The fuzzy clustering algorithm of Equation (6) was applied to both datasets of ambient BC_x_ and meteorology (x = ff or wb), and the total number of clusters sought was varied between four and seven clusters—*p* = 3–6 in Equation (6), respectively. Then, we inspected the time variability of the resulting spatiotemporal patterns (i.e., fuzzy clusters). Based on the similarities in temporal and spatial variability, we identified the major sources contributing to ambient BC_x_ concentrations. Below we discuss the results for both BC components.

#### 3.2.1. Results for BC_ff_

Upon inspection of the different FUSTA results for this BC fraction, the contributions from Santiago’s urban plume arriving at the monitoring site and the noisy cluster contributions were identified by their distinctive upwind locations—W-SW and SSE, respectively (see Figure A4, Figure A5 and Figure 6, below). A residential heating and cooking contribution (RHC) was identified because it is highest overnight when temperatures are lowest—in the winter season. Then, the rest of the contributions are traffic sources located in different directions upwind of the monitoring site. Since the only contribution that vanishes in winter is Santiago’s urban plume, we conclude that all other contributions are local, and they arrive at the monitoring site from different upwind directions and under different combinations of air temperature and pressure (Figure A4, Figure A5 and Figure 6). Table 3 summarizes the mean source contribution estimated in each case. A small variability in major source contribution estimates is observed in these results.

Hence, for simplicity’s sake, we chose the lowest number of fuzzy clusters (5) that apportion all major BC_ff_ sources at play. Figure 5 and Figure 6 display the temporal and spatial variability of those clusters, respectively, and Table 4 provides a statistical summary of cluster contributions; additional bivariate plots for BC_ff_ are presented in Appendix C (Figure A4 and Figure A5). Below we discuss the features of this five-cluster solution.

Cluster 1 contributions rise in the daylight hours and are zero overnight; these contributions increase in the austral summer season and decrease over weekends. Since they come from W/SW directions, this fuzzy cluster is identified as Santiago’s urban plume reaching the monitoring site as anabatic winds develop during daylight. Contributions of this source are highest when temperatures and wind speed increase in the summer season; this air quality feature of the eastern side of Santiago has been described before [36,46]. On average, this source contributes 14% of the total BC_ff_. Notice that in Table 4, this contribution has more than 25% of hours with zero contribution, which corresponds to overnight conditions.

Cluster 2 contributions increase in the evening hours and reach a maximum before sunrise (when the mixing layer is lowest), are highest in the winter season and low temperatures (Figure A4); they arrive at the monitoring site from different directions (NW–NE) which agree with the highest population density surrounding this site. We identify this source contribution as residential heating and cooking (RHC) sources that use compressed natural gas and liquified petroleum gas as fuels. They contribute on average with 13% of total BC_ff_.

Clusters 3 and 4 rise in the morning, peak in the evening, and decrease until dawn; since they also decrease over weekends, we identify those two clusters as local traffic (TRF) sources. Together they account for 66% of total BC_ff_. These two clusters are resolved by the algorithm because they have different seasonality; cluster 4 contributions are higher when temperatures are lower and winds weaker, so this cluster has the highest seasonality of all.

Cluster 5 includes all sources whose spatiotemporal patterns are intermittent, so they are identified as local combustion sources that peak around 1 pm in winter, most likely associated with residential cooking and heating. On average, this source contributes the least to total BC_ff_, with 7%.

#### 3.2.2. Results for BC_wb_

We applied the same criteria to identify the contributions of different FUSTA solutions for the BC_wb_ fraction. Thus, we identified Santiago’s urban plume and noise contributions by their distinctive upwind locations—W-SW and SSE, respectively (Figure A6, Figure A7 and Figure 8, below). Once again, only the urban plume contribution vanishes in the winter season—when a low thermal inversion layer blocks air masses from the lower valley from reaching the monitoring site—so the rest of the sources must be local ones. Table 5 shows the mean source contribution estimated for each source as the number of clusters increases. Again, a small variability in major source contribution estimates is observed in these results.

Again, for simplicity, we have chosen to present the results for four (total) clusters in this case. The results are shown below in Figure 7 and Figure 8 and Table 6; additional bivariate plots for BC_wb_ are presented in Appendix C (Figure A6 and Figure A7). Below we discuss the features of this four-cluster solution.

Clusters 1 and 2 have similar diel profiles with peaks in the evening hours but have different seasonality: cluster 1 contributions peak with ambient temperatures lower than 15 °C (Figure A6) and have a stronger seasonality, while cluster 2 contributions have no clear seasonality pattern and are associated to ambient temperatures above 10 °C (Figure A6). We identify these two clusters as local wood-burning sources; the combined contribution is 75% of total BC_wb_.

Cluster 3 contributions rise in the afternoon and are zero overnight; they peak in the summer season, with high temperatures (Figure A6) and come from W-SW directions. Hence, this is Santiago’s urban plume reaching the monitoring site, and this source contributes 14% of the total BC_wb_.

Cluster 4 contributions come from S-SE directions and peak in winter around 1 pm, with no weekly seasonality. These contributions come under different synoptic conditions of low and high pressure (Figure A7); they likely come from residential cooking and heating and correspond to 11% of total BC_wb_. In this regard, this noise cluster has a similar spatiotemporal pattern as the noise cluster found for BC_ff_; this means the residential sources S-SE of the monitoring site contributes to both BC fractions.

### 3.3. Source Apportionment of BC_ff_ and BC_wb_

The fuzzy clustering methodology (FUSTA) generates a source apportionment of BC at the monitoring site. The following figures show the daily contributions of the different sources resolved in this work; Appendix D presents results for hourly contributions.

Figure 9 and Figure 10 show the daily source contributions for BC_ff_ and BC_wb_, respectively. Local traffic contributions dominate BC_ff,_ and local wood-burning sources dominate BC_wb_. Nonetheless, the noisy source contributions have the largest hourly spikes (Figure A8 and Figure A9). Notice that the urban plume contributions are minimum in wintertime when the mixing layer over the city reaches minimum values [47], blocking air masses from arriving at the monitoring site. The urban plume contribution shows a rise towards the end of 2020, associated with less stringent lockdowns therein and a consequent increase in traffic activity levels [33].

## 4. Discussion

This is the first report of BC source apportionment conducted in Santiago, Chile that includes the effects of lockdowns brought on by the SARS-CoV-2 pandemics. Hence, the results reported here may be considered a baseline for future studies.

The total BC reductions associated with Santiago 2020 lockdowns—from 40% to 80%—are like the ones estimated for other cities worldwide: Delhi, India [25], up to 78%; Kigali, Rwanda [29], 59%; Sommerville, MA, USA [28], 22–56%; Wuhan, China [31], 39%. One limitation of our estimated reduction is that the baseline is not 2019 but 2015; since ambient PM_2.5_ has been steadily decreasing in Santiago for the period 2015–2020 [48], this means our estimates are upper bounds (in magnitude) of 2019–2020 BC reductions.

Regarding BC source apportionment during lockdown conditions, BC_ff_ is dominant all year long, between 82 and 86% of total BC in Santiago. This is higher than in other cities during lockdowns: 70% in Ahmedabad, India [30], 60–86% in Wuhan, China [31], 51–69% in Delhi, India [25], 50% in Kiwali, Rwanda [29]. We ascribe this to the mild, Mediterranean climate of Santiago, the large fleet of motor vehicles therein and the lower proportion of wood-burning emissions as compared to the above cities.

The non-local contributions coming from the greater Santiago metropolitan area are associated with the development of anabatic winds during daylight hours, so these contributions are zero overnight; the spatial and temporal plots (Figure 5, Figure 6, Figure 7 and Figure 8) show that FUSTA methodology separates this contribution from the local sources. This novel approach circumvents the use of an air quality model to estimate how much BC_ff_ (or BC_wb_) originates locally or is transported from upwind urban sources. In addition, the noisy fuzzy cluster concept handles intermittent sources arriving at the monitoring site, which are local emissions from residential cooking and heating; these are resolved from the other local sources because their spatiotemporal patterns are different. This split of local, non-local and intermittent contributions to ambient BC_ff_ and BC_wb_ concentrations will facilitate further air quality modeling studies for these ambient combustion tracer particles.

## 5. Conclusions

A 2020 baseline of ambient BC_ff_ and BC_wb_ concentrations has been compiled for Santiago, Chile, during the SARS-CoV-2 lockdown periods, at an urban site located on the east border of the city. BC_ff_ is the dominant contribution all year long, accounting for more than 80% of total BC. During the more restrictive lockdowns, total BC decreased by ~80% compared with a 2015 ambient BC campaign in the same part of the city; likewise, when lockdowns were relaxed, the decrease in total BC reached ~40% on the same comparison basis.

A new methodology to resolve local and non-local BC sources has been developed. This new methodology is based on a fuzzy clustering of ambient observations of BC and four meteorological variables: wind speed and direction, temperature, and pressure. This new methodology (named FUSTA) can resolve different spatiotemporal patterns (i.e., fuzzy clusters) of ambient BC, which arise from different BC sources contributing to ambient BC concentrations at the monitoring site. The methodology resolves, for instance, the arrival of Santiago’s urban plume to the monitoring site due to the daylight anabatic wind regime in Santiago’s basin. Besides, the methodology also handles intermittent sources like residential heating and cooking, especially in the winter season.

The application of FUSTA methodology to ambient BC_ff_ and BC_wb_ concentrations has led to the result that local sources are dominant in both BC fractions: traffic and wood burning sources, respectively, with 66% and 75%, respectively. The contributions from Santiago’s urban plume arriving at the monitoring site increased towards the end of the year when lockdowns were relaxed; on average, this contribution reached 14% of BC_ff_ and BC_wb_ concentrations. Intermittent residential heating and cooking sources contribute to 7% and 11% of BC_ff_ and BC_wb_ concentrations, respectively. When these intermittent contributions are added to the regular spatiotemporal patterns (clusters), the total contribution of local residential heating and cooking sources reaches up to 20% and 86% for BC_ff_ and BC_wb_ concentrations, respectively.

## Figures and Tables

**Figure 1 ijerph-19-17064-f001:**
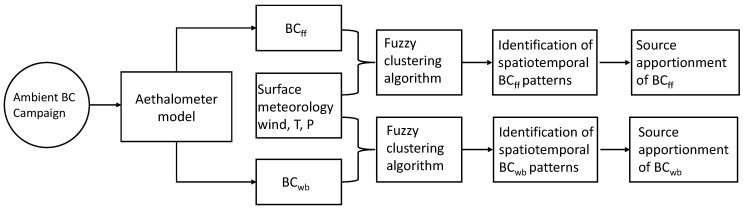
Workflow of the methodology.

**Figure 2 ijerph-19-17064-f002:**
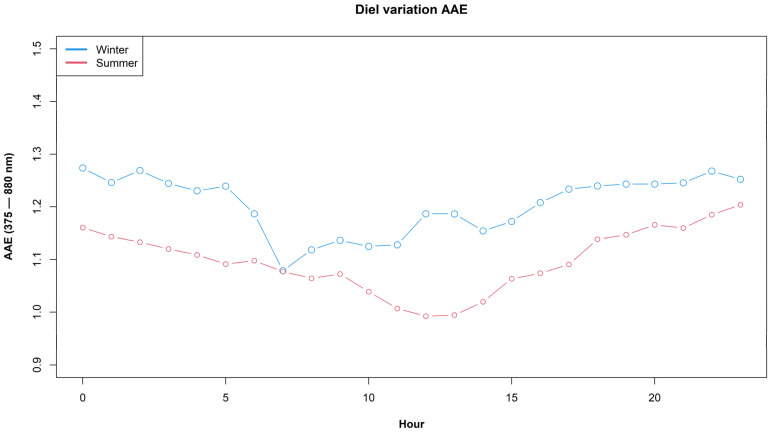
Diel profiles of AAE for (austral) summer and winter months.

**Figure 3 ijerph-19-17064-f003:**
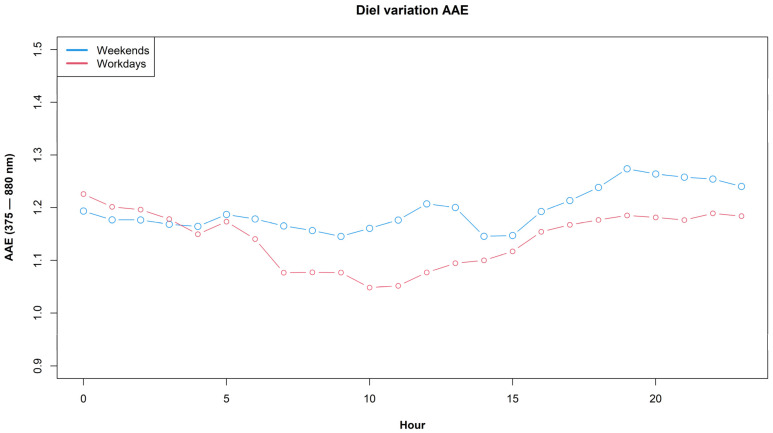
Diel profiles of AAE for workdays and weekends.

**Figure 4 ijerph-19-17064-f004:**
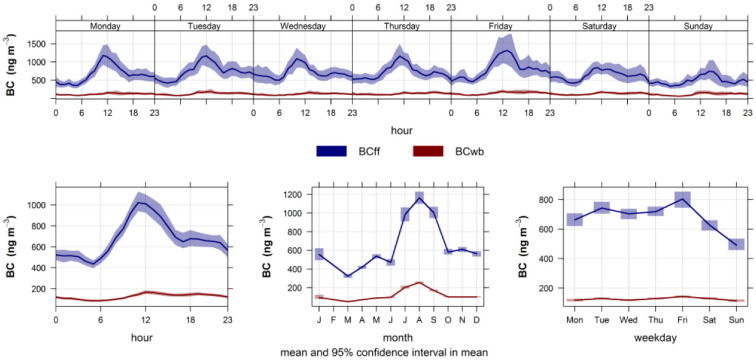
Time variability plot for BC_ff_ and BC_wb_ contributions.

**Figure 5 ijerph-19-17064-f005:**
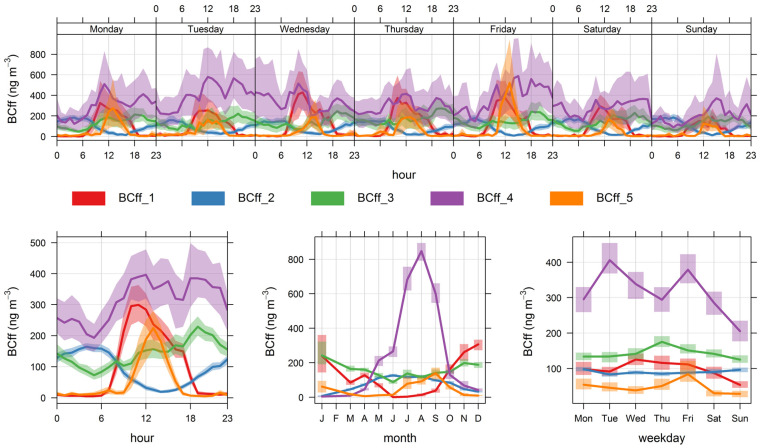
Time variability plot for the five fuzzy clusters’ contributions to BC_ff_.

**Figure 6 ijerph-19-17064-f006:**
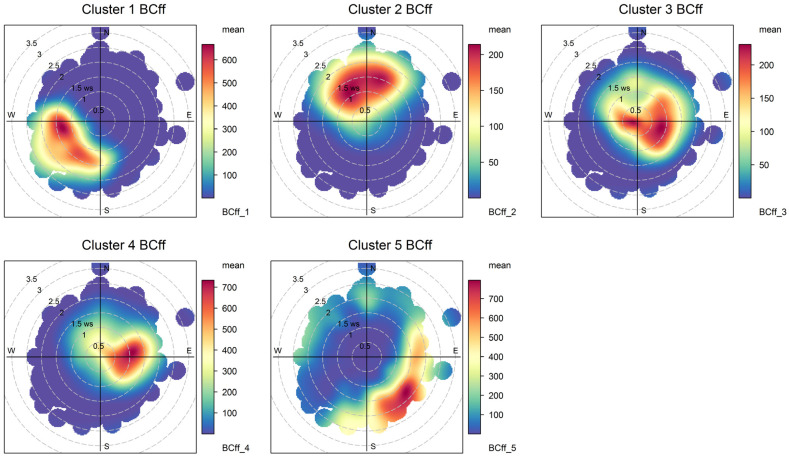
Bivariate plots for the five fuzzy clusters’ contributions to BC_ff_.

**Figure 7 ijerph-19-17064-f007:**
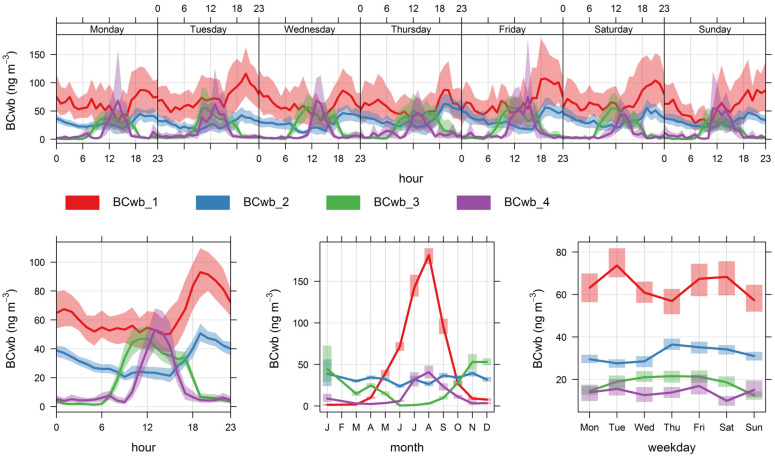
Time variability plot for the four fuzzy clusters’ contributions to BC_wb_.

**Figure 8 ijerph-19-17064-f008:**
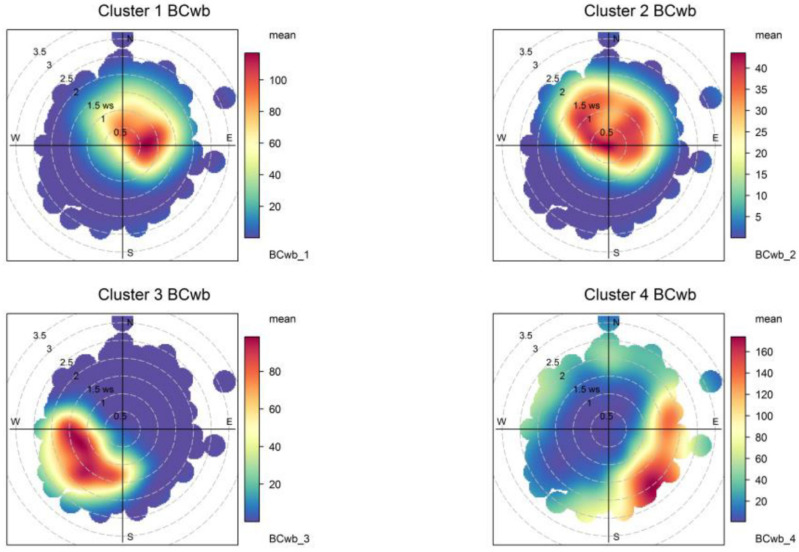
Bivariate plots for the four fuzzy clusters’ contributions to BC_wb_.

**Figure 9 ijerph-19-17064-f009:**
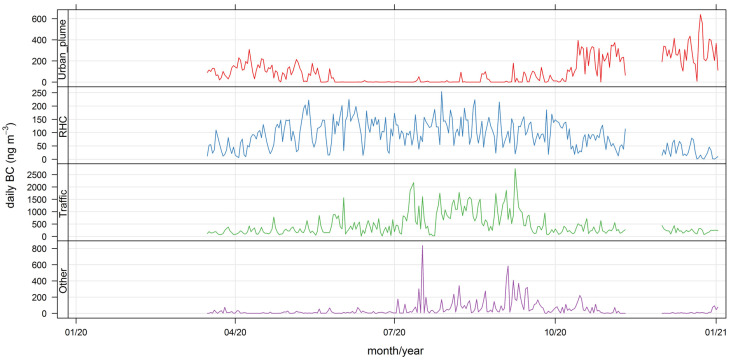
Time plot of daily source contributions to BC_ff_.

**Figure 10 ijerph-19-17064-f010:**
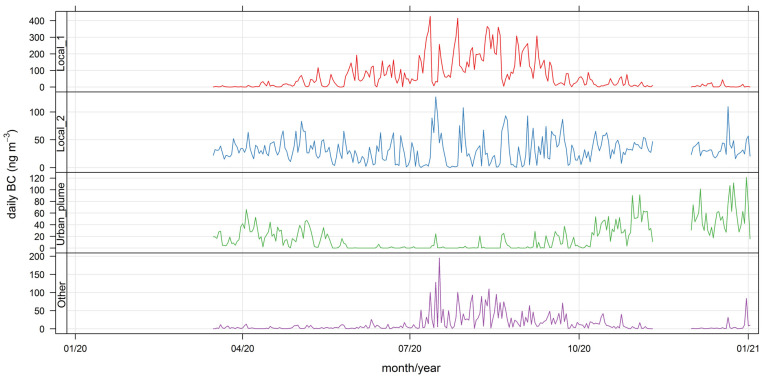
Time plot of daily source contributions to BC_wb_.

**Table 1 ijerph-19-17064-t001:** Statistical summary of campaign results ^1^.

Statistic	AAE (-)	BC_ff_ (ng/m^3^)	BC_wb_ (ng/m^3^)	BC (ng/m^3^)
Minimum	0.002	0.62	0.04	0.11
1st quantile	1.036	294	55	357
Median	1.135	488	90	585
Mean	1.160	690	128	809
3rd quartile	1.237	847	156	988
Maximum	3.322	7812	1604	8679

^1^ Negative values are excluded from the statistics (see also Appendix B).

**Table 2 ijerph-19-17064-t002:** Comparison of ambient BC measurements in east Santiago, monthly averages (ng/m^3^).

Month	2015 ^1^	2020 ^3^	Ratio 2020/2015
January	1200		
February	1063		
March	1583	376 ± 50	0.24
April	2233	495 ± 62	0.22
May	2440	627 ± 100	0.26
June	2877	533 ± 97	0.19
July	2430	1207 ± 279	0.50
August		1435 ± 196	
September		1137 ± 268	
October		682 ± 93	
November		706 ± 90	
December	1073 ^2^	672 ± 68	0.63

^1^ Data adapted from [24]. ^2^ Data correspond to December 2014. ^3^ Data reported as mean ± 2σ, estimated from daily averages with at least 75% of valid hourly values.

**Table 3 ijerph-19-17064-t003:** Mean source contributions to BC_ff_ (ng/m^3^) for a different choice of total clusters sought.

Total Clusters	Urban Plume	RHC	TRF	Other (Noise)
4	109	unresolved	505	80
5	98	90	458	48
6	91	78	486	40
7	87	54	528	27

**Table 4 ijerph-19-17064-t004:** Statistical summary of hourly BC_ff_ source contributions (ng/m^3^) for a 5-cluster solution.

Statistic	Cluster 1	Cluster 2	Cluster 3	Cluster 4	Cluster 5
Minimum	0.0	0.0	0.0	0.0	0.15
1st quantile	0.0	5.2	4.5	0.7	0.63
Median	0.22	51.5	45.0	24.2	1.86
Mean	98.0	90.4	143	315	47.7
3rd quartile	17.4	150.8	215	313	9.4
Maximum	2313	641	1695	5304	6296

**Table 5 ijerph-19-17064-t005:** Mean source contributions to BC_wb_ (ng/m^3^) for a different choice of total clusters sought.

Total Clusters	Urban Plume	Local Wood Burning	Other Local (Noise)
4	18	96	14
5	17	103	8
6	15	106	7
7	14	109	5

**Table 6 ijerph-19-17064-t006:** Statistical summary of BC_wb_ source contributions (ng/m^3^).

Statistic	Cluster 1	Cluster 2	Cluster 3	Cluster 4
Minimum	0.00	0.0	0.00	0.04
1st quantile	0.34	2.37	0.00	0.24
Median	13.1	19.1	0.12	0.73
Mean	64.0	31.8	18.4	14.0
3rd quartile	82.6	49.7	7.9	4.1
Maximum	873	265	387	1506

## Data Availability

Data are provided as Appendix A (see above).

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
