# Peer review of "Spatiotemporal Analysis of Black Carbon Sources: Case of Santiago, Chile, under SARS-CoV-2 Lockdowns"

_ijerph, 2022, doi:10.3390/ijerph192417064_

Round 1
Reviewer 1 Report
A spatiotemporal amalysis of black carbon emissions in urban sites using a fuzzy clustering technique to separe BC sources is proposed.
The goals of the research are clear, however in the introduction there is a short discussion on the critical points and shortcomings of the methods of spatiotemporal analysis of local and global BC sources [25, .., 32] that motivated the aimes of this research.
In section 2 it is necessary to add a flow diagram that summarizes the experimented process in a structured way and by steps.
Authors declare that they have used the fuzzy clustering algorithm [40]; however in [40] a toolbox for fuzzy clustering using the R programming language is presented, a specific fuzzy clustering algorithm is not proposed. There are a large number of fuzzy clustering algorithms in the literature. Fuzzy C-means itself, the best known fuzzy clustering algorithm, has many variations. Therefore, authors must specify which fuzzy clustering algorithm is used in the research.
How are the 5 clusters detected? Authors must explain whether the 5 clusters are fixed a priori (as in FUzzy Cmeans) or are determined directly by the cfuzzy clustering algorithm. In the first case they must justify the choice of the 5 clusters, in the second case they must specify which fuzzy clustering method that determines the final number of clusters has been used. In both cases, authors must specify which are the input parameters set in the execution of the fuzzy clustering algorithm and justify their choices.
Author Response
A spatiotemporal analysis of black carbon emissions in urban sites using a fuzzy clustering technique to separate BC sources is proposed.
- a) The goals of the research are clear, however in the introduction there is a short discussion on the critical points and shortcomings of the methods of spatiotemporal analysis of local and global BC sources [25, .., 32] that motivated the aims of this research.
Response: we have edited the introduction section to better explain our research goals.
- b) In section 2 it is necessary to add a flow diagram that summarizes the experimented process in a structured way and by steps.
Response: we have included a flowchart of the whole quantitative analysis performed (Figure 1)
Figure 1. Workflow of the methodology.
- c) Authors declare that they have used the fuzzy clustering algorithm [40]; however in [40] a toolbox for fuzzy clustering using the R programming language is presented, a specific fuzzy clustering algorithm is not proposed. There are a large number of fuzzy clustering algorithms in the literature. Fuzzy C-means itself, the best known fuzzy clustering algorithm, has many variations. Therefore, authors must specify which fuzzy clustering algorithm is used in the research.
Response: We use the routine “FKM.ent.noise” available in that R toolbox (fclust). We clarify this in the manuscript. Besides, the supplementary files contain the database and the macros used to generate fuzzy clustering results and their graphical visualization, so any reader may go through the workflow shown on Figure 1.
- d) How are the 5 clusters detected? Authors must explain whether the 5 clusters are fixed a priori (as in FUzzy Cmeans) or are determined directly by the cfuzzy clustering algorithm. In the first case they must justify the choice of the 5 clusters, in the second case they must specify which fuzzy clustering method that determines the final number of clusters has been used. In both cases, authors must specify which are the input parameters set in the execution of the fuzzy clustering algorithm and justify their choices.
Response: To arrive at 5 (total) clusters for the BCff fraction, we performed the fuzzy clustering algorithm with p= 4,5,6 and 7 (total) clusters and inspected the time variability of the resulting spatiotemporal patterns (i.e., fuzzy clusters). Based on the similarities in time and spatial variation, we identified the urban plume and the noisy cluster contributions by their distinctive locations — SW and SSE, respectively (see Figures 6, B1 and B2). The residential heating and cooking contribution (RHC) is highest overnight when temperatures are lowest (winter season). Then, the rest of the contributions are traffic sources located at different directions upwind the monitoring site. Since the only contribution that vanishes in winter is the urban plume, we conclude that all other contributions are local, and they arrive to the monitoring site from different upwind directions and under different combinations of air temperature and pressure. Table 3 summarizes the mean contribution in each case. A small variability in major source contribution estimates is observed in these results.
Table 3. Mean source contributions to BCff (ng/m3) for different choice of total clusters sought.
Total clusters |
Urban plume |
RHC |
TRF |
Other (noise) |
4 |
109 |
unresolved |
505 |
80 |
5 |
98 |
90 |
458 |
48 |
6 |
91 |
78 |
486 |
40 |
7 |
87 |
54 |
528 |
27 |
Hence, for simplicity’s sake, we chose the lowest number of fuzzy clusters (5) that apportion all major BCff sources. This is clarified in the revised manuscript, at the beginning of section 3.2.1.
Likewise, for the BCwb fraction, we identified the urban plume and noise contributions because these have the very same distinctive upwind locations — SW and SSE, respectively — than the case of BCff (Figures 8, B3 and B4). Once again, only the urban plume contribution vanishes in winter season — when a low thermal inversion layer blocks air masses from the lower valley to reach the monitoring site — so the rest of the sources must be local ones. The following table shows the mean source contribution estimated for each source as the number of clusters increases.
Table 5. Mean source contributions to BCwb (ng/m3) for different choice of total clusters sought.
Total clusters |
Urban plume |
Local wood burning |
Other local (noise) |
4 |
18 |
96 |
14 |
5 |
17 |
103 |
8 |
6 |
15 |
106 |
7 |
7 |
14 |
109 |
5 |
Again, for simplicity, we have chosen to present the results for 4 (total) clusters in this case.
We have provided a digital supplementary file with all data needed to reproduce the analysis presented here and in the revised manuscript.

Reviewer 2 Report
Some of my points that need to be fixed in this revision
-add a summary for the related works in a paragraph or a table to show for the reader what is the difference between your work and other published works in the literature.
-A native speaker should check the language.
-Add numbering to all main sections
-make your contributions clearer
-check the mathematical notations
-As, the work proceeds in a mathematical perspective derivations are not necessary the meaningful contribution with its research findings has to be stated well.
- Please use the complete form of abbreviations in the abstract.
- Your ideas in the introduction section need to be more comprehensive.
- The last paragraph before the end of section one should contain at least the follows The proposed methods clearly what is the main differences between the proposed algorithm and the others How the contributions were done. The problem that has been solved in this research. The datasets that have been used in the research experiments. The general results that you have been got.
-The novelty of the method is limited. The presentation (clarity/structure) is average and the description for the method is not very easy to understand.
Reviewer 3 Report
This work introduces a fuzzy clustering approach as a tool for emission-source analysis of BC pollution during SARS-CoV-2 pandemics. This work contains something interesting and novel. The authors have written the manuscript well and try to show the new sciences on this paper taking some statistical analysis. However, there are some key problems in the paper.
1. The study is only a polluted point study. The statistical analysis needs multi location data. The authors should provide more information and explanation about study data of one site.
2. The fluctuations of BC can be affected by the meteorological factors, such as precipitation and wind speed. Why not analyze the impact of other factors in this paper? The authors needs to strengthen the discussion on the influence of meteorological factors.
3. Authors demonstrate that the novel fuzzy clustering approach can help identify the local and non-local BC pollution. Besides BC, can the novel model be applied to other air pollutants, such as PM2.5? How accurate is this model, compared to other atmospheric transport numerical models?
Round 2
Reviewer 1 Report
Authors have taken all my suggestions into account, inserting a flochart of the experimental process carried out and motivating the choice of cluster number in detail. I consider this paper publishable in the present form.
Reviewer 2 Report
accept
Reviewer 3 Report
The author has made positive responses and modifications to the concerns of the reviewers. In my opinion, this article is suitable for publication.